# Recurrence Patterns after Surgery in Patients with Different Endometriosis Subtypes: A Long-Term Hospital-Based Cohort Study

**DOI:** 10.3390/jcm9020496

**Published:** 2020-02-11

**Authors:** Konstantinos Nirgianakis, Lijuan Ma, Brett McKinnon, Michael D. Mueller

**Affiliations:** 1Department of Gynecology and Gynecological Oncology, Inselspital, Bern University Hospital, University of Bern, Friedbuehlstrasse 19, 3010 Bern, Switzerland; 2Department for BioMedical Research, University of Bern, 3008 Bern, Switzerland

**Keywords:** recurrence, progression, peritoneal endometriosis, endometrioma, deep-infiltrative endometriosis

## Abstract

Recurrence of endometriosis after surgery constitutes a serious challenge. Whether there is an evolution of lesion subtypes with each recurrence and whether certain lesions subtypes tend to recur faster than others is not adequately addressed. Medical records of all patients who underwent surgery for endometriosis between 1997 and 2018 in the Department of Gynecology and Obstetrics, University of Bern, were reviewed. Inclusion criteria was surgically confirmed endometriosis recurrence, defined as a subsequent surgery for endometriosis after a previous complete surgical excision of endometriosis lesions. Three subtypes of endometriosis were defined: superficial peritoneal endometriosis (SUP), ovarian endometrioma (OMA), and deep infiltrating endometriosis (DIE). Time to recurrence and variation in endometriosis subtype between the first and recurrent surgeries were the primary outcome measures. Out of the 322 patients with recurrent surgery that were identified, for 234 of them, the endometriosis subtype at first surgery was confirmed and classified (SUP = 56, OMA = 124, DIE = 54). No statistically significant difference was found for time to recurrence between lesion subtypes. SUP compared to the other groups had a higher possibility of presenting with SUP at recurrence (Odds Ratio (OR): 3.65, 95% confidence interval (CI): 1.74–7.51) and OMA compared to the other groups had a higher possibility of presenting with OMA at recurrence (OR: 3.72, 95% CI: 2.04–6.74). Nevertheless, a large number of SUP patients subsequently presented with OMA (10/56: 17.9%) or DIE (27/56: 48.2%) lesions at recurrence. Similarly, a large number of OMA patients subsequently presented with DIE (49/124: 39.5%) lesions at recurrence. In conclusion, although SUP and OMA patients compared to the others are more likely to present with the same subtype at recurrence, increasing lesion subtype severity occurs in a substantial proportion of patients. Time to recurrence is independent from the lesion subtype at first surgery.

## 1. Introduction

Endometriosis, characterized by the growth of endometrial-like tissue outside the uterine cavity, is a highly prevalent gynecological disorder of reproductive-aged women worldwide [1,2,3]. It is a significantly heterogeneous disease, both in phenotype and clinical outcomes that can lead to a significant reduction in quality of life and work productivity [4,5]. Recommended treatments for endometriosis are either hormonal-based therapy or laparoscopic surgical excision depending on response and tolerability to medical treatment, as well as family planning.

Laparoscopic surgery is associated with decreased overall pain, both at 6 and 12 months after surgery [6]. However, despite complete removal of endometriotic tissue, a high proportion of patients will require additional surgery due to endometriosis recurrence. In a recent UK population-based report, 48% of patients with endometriosis received surgical treatment. Approximately one-fifth of these patients required further surgical treatment, within three years of the index procedure [7]. Other studies have reported total recurrence rates of 21.5% and 40%–50% at two and five years, respectively [8,9].

Endometriosis lesions are a heterogenous group of lesions that are currently split into three subtypes based on the location and infiltration depth: superficial peritoneal endometriosis (SUP), ovarian endometrioma (OMA), and deep infiltrating endometriosis (DIE) [10,11]. Although patients with SUP may suffer from pelvic pain, OMA and DIE generally cause heavier symptoms, have more serious long-term complications, are more difficult to manage [12,13,14,15,16], and thus considered as more severe endometriosis subtypes. Whether there is an evolution of lesion subtypes over each recurrence and whether certain lesions subtypes recur faster than others is not adequately addressed. Research up to now has been scant, limited to adolescence and with contradictory results [17,18,19,20,21].

The purpose of this study, therefore, was to characterize the lesion subtypes in first and subsequent surgeries, examine their evolution and compare the time required for subsequent surgery based on the initial lesion subtype.

## 2. Materials and Methods

The study was prepared according to the “Strengthening the reporting of observational studies in epidemiology” guidelines [22] and was institution review board approved (no. 2017-00952). The electronic medical records were searched for all patients who underwent at least one laparoscopic surgery for endometriosis in the Department of Gynecology and Obstetrics, University of Bern (between January 1997 and October 2018). The initial search for inclusion criteria was performed by one researcher (L.M.) and the medical records identified for inclusion were reviewed by two independent researchers (K.N.) (L.M). Only patients with more than one surgery for endometriosis were included, while surgeries in external hospitals were not excluded. For all surgeries, either visual or histological confirmation of endometriosis was required for inclusion. Unavailable surgical reports or undefined surgical technique, incomplete excision of endometriosis lesions, and diagnostic surgeries made up the exclusion criteria. Recurrence was defined as subsequent surgery for endometriosis after a previous, complete surgical excision of endometriosis. Recurrence of endometriosis symptoms or recurrence of endometriosis based on clinical suspicion or imaging was not evaluated.

Surgical data, histological results and time to recurrence were collected and analyzed retrospectively. The classification of endometriosis subtype was performed according to the most severe endometriotic lesion identified [23]. As a result, DIE with concomitant OMA and/or SUP was classified as DIE. OMA with concomitant SUP was classified as OMA.

### 2.1. Surgical Technique

The standardized laparoscopic surgical technique for DIE performed in our clinic has been described previously [24]. SUP was treated by excision via monopolar needle or scissors. OMA was treated by the striping technique.

### 2.2. Statistical Analysis

Median values and range, or mean values and standard deviation (SD) were calculated for continuous variables and percentages for the qualitative variables. The time to recurrence was assessed according to the Kaplan-Meier life-table analysis. A log-rank test was used to compare the recurrence rates between groups. An ordinary one-way ANOVA and Kruskal–Wallis test were used to compare continuous parametric and nonparametric variables, respectively. Fisher’s exact test was used to compare the proportion of endometriosis subtypes at each recurrence and to determine whether lesion subtype was more or less severe. Significance was set at a *p* value of <0.05. Statistical analysis was carried out with GraphPad Prism version 8.0.1 (GraphPad Software, San Diego, CA 92108, USA).

## 3. Results

### 3.1. Patient Characteristics

Among 1332 patients with surgically diagnosed endometriosis, 322 satisfied both the inclusion and exclusion criteria. For 234 (72.7%), the endometriosis subtype at first surgery was verified from the surgical report. For the remaining 88 patients, the endometriosis subtype was unclear. The patients’ characteristics recorded at the initial surgery are summarized in Table 1. 

### 3.2. Time to Recurrence

The median time to first recurrent surgery, irrespective of lesion subtype, was 32 months (5–244 months) (Appendix A). For patients who underwent a second recurrent surgery, it was performed after an additional 35 months (5–222 months). Surgery for the third and fourth recurrence were performed after 30 (6–160 months) and 34 (5–90 months) months, respectively. The times between surgeries for each recurrence were not statistically significantly different (Appendix A).

For patients categorized in the SUP group, based on their first surgery for endometriosis, the median time to their first recurrence was 30.5 (5–216) months. For patients categorized in the OMA group, this time was 30 (6–244) months and for patients categorized in the DIE group this was 36 (4–141) months. The time to recurrence for each lesion subtype was not statistically significantly different (Figure 1).

### 3.3. Recurrent Endometriosis Subtype, Based on Subtype at First Surgery

Patients that had a SUP at the first surgery were more likely to present again with a SUP at subsequent surgery (17/56: 30.4%), compared to women that originally had an OMA (10/124: 8.1%), or women that originally had a DIE (9/54: 16.7%). This difference was statistically significant (OR: 3.65, 95% CI: 1.74–7.51; *p* = 0.001). Similarly, patients that had an OMA at first surgery were more likely, to have an OMA (58/124: 46.8%) at subsequent surgery compared to women that originally had a SUP (10/56: 17.9%), or women that had a DIE (11/54: 20.6%). This difference was statistically significant (OR: 3.72, 95% CI: 2.04–6.74; *p* < 0.0001). Patients that initially presented with DIE showed a trend to also subsequently present with DIE (29/54: 53.7%) at the next surgery. However, compared to the other groups, it was not statistically significantly higher, reflecting the high percentage of patients from the other groups that had DIE at subsequent surgeries (Table 2 and Figure 2).

### 3.4. Evolution of Endometriosis Subtypes over Recurrent Surgeries

Interestingly, although the above results suggest the lesion subtype present at the first surgery is a good indication of the lesion to expect at the recurrent surgery, the results also show that a substantial proportion of patients with initially SUP or OMA lesions progress to a more severe subtype at recurrence. Of the women initially presenting with SUP, 66.1% returned for recurrent surgery with either an OMA (10/56: 17.9%) or DIE (27/56: 48.2%), which was statistically significant more when both of these subtypes were combined vs. SUP at recurrence (17/56: 30.4%), (*p* = 0.0295). Similarly, of the women initially diagnosed with an OMA, 39.5% (49/124) returned for recurrent surgery with DIE, whereas only 8.1% (10/124) with the less severe SUP (Table 2 and Figure 2), which was also statistically significant based on a Fisher exact test of more vs. less severe lesions (*p* < 0.0001).

There were two patients (3.6%) in the initially SUP group that had a concomitant hysterectomy at recurrent surgery. One patient (1.8%) underwent hysterectomy during the first surgery, but still required a subsequent surgery for a recurrence of OMA. Six patients (5%) in the initially OMA group underwent a concomitant hysterectomy at recurrent surgery. Three patients (2.4%) underwent a hysterectomy during the first surgery, all three of which had an OMA lesion at subsequent surgery, one of which also had a DIE lesion. Eight patients (15.4%) in the DIE group underwent a concomitant hysterectomy at the second surgery. Two patients (3.7%) underwent hysterectomy during the first surgery and both subsequently presented with DIE at the next surgery; one of which was combined with an OMA. Concomitant hysterectomy at recurrence was significantly more common in the DIE compared to the SUP or OMA group (OR: 3.00, 95% CI: 1.11–7.73; *p* = 0.007).

The majority of the OMA identified at subsequent surgeries occurred on the same ovary as the initial surgery (Table 3 (a)). Similarly, the majority of the patients that presented with DIE lesions at first surgery were most likely to have recurrent lesions in the same area at the subsequent surgery (Table 3 (b)).

Of the 322 patients that underwent at least two surgeries for endometriosis, 128 (39.8%) had an additional 3rd surgery and 48 (14.9%) a 4th surgery. In these patients, we observed a similar trend with a high proportion of patients presenting with more severe subtypes and in particular DIE lesions at subsequent surgery. The data are presented in the Appendix A (2nd to 3rd surgery) and Appendix A (3rd to 4th surgery). The discrepancy between the above-referred total numbers of recurrences and the numbers in the Appendix A is due to some patients for which the lesion subtypes could not be classified, thus not included in the tables. Due to the limited sample numbers, a statistical analysis was not performed.

## 4. Discussion

In the present study, we demonstrate that the time to first recurrent endometriosis surgery is independent from the endometriosis subtype observed at the initial surgery. Moreover, at subsequent surgery the endometriosis subtype observed is likely to be the same subtype observed previously. Interestingly, however, there is a high percentage of patients that present with more severe lesion subtypes, particularly DIE. The trend towards more severe endometriosis subtypes in these patients implies disease progression may occur overtime irrespective of surgical removal.

To the best of our knowledge, this is the first study to compare the time to recurrence between different endometriosis subtypes. The median time to first recurrence for all women with endometriosis was 31 months, similar to the 30 months reported by Liu et al. [25].

It is important to note that our study only included patients who had recurrent endometriosis lesions confirmed through a second surgery. We have specifically selected this cohort because (1) we believe that given a long enough follow up there is significant potential for all women with endometriosis to recur, and thus it becomes difficult to define a non-recurrence group, particularly if surgery is required to confirm the diagnosis and, (2) by including women that have had an initial, complete excision of endometriosis we can confirm that at this point in time these women were devoid of macroscopic endometriosis lesions. This study does not report on women who did not require subsequent surgical intervention, thus it does not describe the likelihood of all endometriosis to recur, but rather only the time to surgical intervention for the group of women with endometriosis that have recurrence significant enough to require additional surgical intervention. Further studies that examine whether there are differences in recurrence in all patients, i.e., those that do not require surgical intervention would be interesting, but challenging to design and perform.

Whether endometriosis represents a progressive disease that worsens over time is not resolved, but is attracting more attention. A three-year prospective study suggested that endometriosis is a progressive disease [13], which was supported by a review of adolescence endometriosis [17], with an additional study showing development from peritoneal to ovarian endometriosis, including uterosacral ligament lesions during a two to five year follow up [19]. On the contrary, an analysis of randomized control studies (RCT) on adolescents showed 71% of the patients without endometriosis excision did not progress [18]. However, a single RCT contributed the majority of cases to this analysis with a short follow up of only four to six months [26]. The findings of our study agree with the suggestion of progression and extend them to a broader population with a longer follow up.

OMA was the most common endometriosis subtype that was observed in this group of patients (53.4%). With OMA being the easiest type of endometriosis to diagnose, it seems plausible that they are more usually treated via surgery. If the higher incidence of recurrence is solely due to an observation bias, or that OMA is more likely to recur cannot be answered by the current study. A recent randomized controlled study evaluating levonorgestrel-IUD reported 25%–37.5% OMA recurrence [27]. However, recurrent surgery on the ovary is related to ovarian reserve damage [28] and recurrent surgery for endometriosis in general is related to stress, complications, as well as personal and social costs. Therefore, caution is required firstly to adhere to the guidelines on the indications for endometriosis surgery and surgical technique and secondly to decrease the risk of recurrence via hormonal suppressive therapy [29,30,31,32].

The underlying pathogenesis of endometriosis recurrence is unclear. If recurrence derives from residual endometriotic cells that remain after surgery, or from de novo lesions is a matter of debate [8]. It has been reported that DIE lesions reappeared at subsequent surgery in the same area of the pelvis as at the previous surgery [33], possibly due to the high number of incomplete surgeries included in the study. In another study, 50 out of 62 (80.6%) patients with recurrent endometriomas had recurrence on the treated ovary [34]. Our study also shows that the majority of recurrent endometriomas were on the same ovary, which could indicate residual lesions. However, lesions on the contralateral ovary, or other areas were also observed indicating that de novo lesion development is possible especially since some recurrences were documented after a long nascent period. Another possibility could be that recurrent lesions at different locations than the initial lesion could occur by endometriotic tissue dissemination during the first surgery.

An important limiting factor in endometriosis research is that although endometriosis recurrence can be well defined within a retrospective study, identifying and confirming non-recurrence is impossible. Thus, no non-recurrent group of patients was included in our study. Many patients may not undergo a subsequent surgery for endometriosis, choosing to tolerate endometriosis-associated symptoms, or to treat the symptoms medically. A detailed follow-up including this information is perhaps possible within a prospective study with long enough follow-up, since many recurrences occur after 48 months or even longer. However, due to the difficulty of an accurate non-surgical diagnosis of endometriosis the classification of a patient as non-recurrent would be challenging, even in such a prospective follow-up study.

Based on the surgical protocols, we assumed in each case that the endometriotic lesions were completely excised. It is however possible that this was not always the case, and thus, some subsequent surgeries could be the result of endometriosis persistence instead of endometriosis recurrence. The same applies to the endometriosis subtype with some misdiagnosed at the initial surgery. Moreover, we cannot exclude that selection bias due to the tertiary nature of our clinic was in part a reason for the high proportion of patients with endometriosis progression since less severe endometriosis subtypes may have been surgically treated in other hospitals. However, the tendency of endometriosis progression was observed also on the patients exclusively treated in our clinic. Finally, we lacked reliable data on postoperative hormonal medication. Although there is a consistency of prescription within the single clinic, it is impossible to confirm compliance. We could assume, however, that since the time to recurrence was not statistically significantly different between groups, there was also no significant difference in the hormonal medication used.

The main advantages of the current study should also be mentioned. It provides real world data and contrary to available studies, a very long follow-up, often including the whole reproductive time, traversing many recurrent surgeries per patient. Moreover, recurrences are well defined and described by laparoscopy, not by imaging alone, adding to the accuracy of the observations. Finally, the inclusion of all hospital-based recurrent patients provides more generalizability compared to the population included in available RCTs primarily designed for other purposes.

The results of the study will help clinicians to better comprehend the evolution of endometriosis in recurrent surgeries and ultimately provide valuable information for patient counselling, especially after surgery.

## Figures and Tables

**Figure 1 jcm-09-00496-f001:**
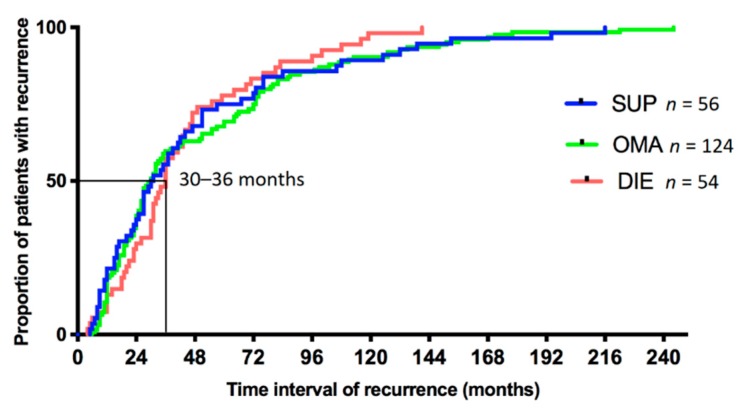
Time to first recurrent surgery according to the initial type of endometriosis. Legend: The time to recurrence is illustrated in different colors according to the endometriosis lesion subtype at initial surgery. No statistically significant difference was observed. Abbreviations: SUP, superficial peritoneal endometriosis; OMA, ovarian endometrioma; DIE, deep infiltrating endometriosis.

**Figure 2 jcm-09-00496-f002:**
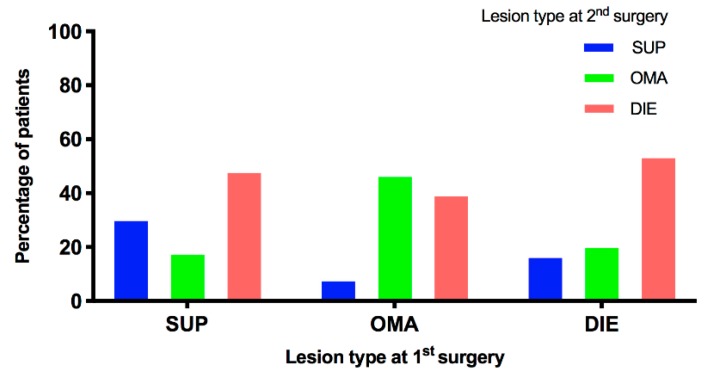
The evolution of SUP, OMA, and DIE at first recurrence. Legend: Graphical illustration of Table 2. The evolution of the lesions from the first to recurrent surgery is presented. Each group of patients is split into 3 colored columns with each color representing a certain lesion subtype at recurrent surgery. Abbreviations: SUP, superficial peritoneal endometriosis; OMA, ovarian endometrioma; DIE, deep infiltrating endometriosis.

**Table 1 jcm-09-00496-t001:** Patient’s characteristics according to the endometriosis subtype at first surgery.

	First Surgery	SUP	OMA	DIE	Unknown	*p*
Characteristics		(*n* = 56)	(*n* = 124)	(*n* = 54)	(*n* = 88)	
Age (y ± SD)		27.7 ± 6.4	29.4 ± 5.3	30.1 ± 5.0	29.4 ± 6.6	ns
Median time to second surgery (min-max, months)		30.5 (5–216)	30 (6–244)	36 (4–141)	33.5 (5–190)	ns
First surgery in external hospital		43 (76.8%)	109 (87.9%)	33 (61.1%)	85 (96.6%)	<0.001 *
Second surgery in external hospital		12 (21.4%)	40 (32.3%)	12 (22.2%)	33 (37.5%)	ns
One recurrence		100%	100%	100%	100%	n/a
Two recurrences		18 (32.1%)	52 (41.9%)	18 (33.3%)	44 (50.0%)	ns
Three recurrences		4 (7.1%)	16 (12.9%)	8 (14.8%)	22 (25.0%)	ns
Four recurrences		2 (3.6%)	4 (3.2%)	4 (7.4%)	13 (14.8%)	ns
Five recurrences		0	0	1 (1.9%)	3 (3.4%)	ns

***** Statistical comparison performed by Chi-square Test. Abbreviations: SUP, superficial peritoneal endometriosis; OMA, ovarian endometrioma; DIE, deep infiltrating endometriosis; y, years; SD, standard deviation; ns, not significant.

**Table 2 jcm-09-00496-t002:** Evolution of endometriosis from first to recurrent surgery.

	First Surgery	SUP (N = 56) Median Time to Recurrence (Min–Max)	*p* OR (95% CI) ^1^	OMA (N = 124) Median Time to Recurrence (Min–Max)	*p* OR (95% CI) ^2^	DIE (N = 54) Median Time to Recurrence (Min–Max)	*p* OR (95% CI) ^3^
Recurrent Surgery							
**SUP**		17 (30.4%)	0.001	10 (8.1%)	0.0011	9 (16.7%)	ns
		30 (9–194)	3.65	28 (7–244)	0.28	31 (5–116)	
			(1.74, 7.51)		(0.14, 0.62)		
**OMA**		10 (17.9%) 71.5 (6–216)	0.0036 0.34 (0.17, 0.73)	58 (46.8%) 27 (6–222)	<0.0001 3.72 (2.04, 6.74)	11 (20.4%) 36 (6–141)	0.021 0.42 (0.2, 0.87)
**DIE**		27 (48.2%)	ns	49 (39.5%)	ns	29 (53.7%)	ns
		27 (5–139)		51 (8–135)		39 (18–119)	
Unknown subtype		2 (3.6%)	ns	5 (4.0%)	ns	5 (9.3%)	ns
		31.5 (12.51)		30 (12–156)		24 (4–31)	

In each cell, the number of cases evolving into SUP, OMA, DIE, or unknown lesion subtype at recurrent surgery and their percentage is given. ^1^: The ORs in this column reflect the possibility of a patient with initially SUP lesions compared to a patient with initially non-SUP lesions (OMA and DIE) to present a certain endometriosis lesion at recurrent surgery. The bold numbers represent the statistically significant higher possibility of SUP patients, compared to non-SUP patients to present with SUP lesions again, in the absence of OMA and DIE at recurrent surgery. ^2^: The ORs in this column reflect the possibility of a patient with initially OMA lesions compared to a patient with initially non-OMA lesions (SUP and DIE) to present a certain endometriosis lesion at recurrent surgery. The bold marked numbers represent the statistically significant higher possibility of OMA patients, compared to non-OMA patients to present with OMA lesions again in the absence of DIE at recurrent surgery. ^3^: The ORs in this column reflect the possibility of a patient with initially DIE lesions compared to a patient with initially non-DIE lesions (SUP and OMA) to present SUP, OMA or DIE at recurrent surgery. Abbreviations: SUP, superficial peritoneal endometriosis; OMA, ovarian endometrioma; DIE, deep infiltrating endometriosis; ns, not significant; Min-Max, Minimum-Maximum; OR (95% CI), Odds Ratio (95% Confidence Interval).

**Table 3 jcm-09-00496-t003:** (**a**) Location analysis of OMA at first surgical recurrence. (**b**) Location analysis of DIE at recurrent surgery.

**(a)**									
		**Location of OMA at 1st Surgery**	**Bilateral** ***n* = 13 (22.4%)**			**Unilateral Left** ***n* = 21 (36.2%)**		**Unilateral Right** ***n* = 14 (24.1%)**	**Unknown** ***n* = 10 (17.2%)**
**Location of OMA at 2nd Surgery**									
Bilateral			6 (46.2%)			9 (42.9%)		4 (28.6%)	4 (40.0%)
Unilateral left			3 (23.1%)			8 (38.1%)		3 (21.4%)	1 (10.0%)
Unilateral right			4 (30.8%)			4 (19.0%)		7 (50.0%)	3 (30.0%)
Unknown			0 (0%)			0 (0%)		0 (0%)	2 (20.0%)
**(b)**									
	**First Surgery**		**DIE**					**SUP** ***n* = 27**	**OMA** ***n* = 49**
			**Uterosacral Ligament** ***n* = 3/29**	**Vagina** ***n* = 11/29**	**Intestine** ***n* = 10/29**	**Bladder** ***n* = 1/29**	**Others *** ***n* = 4/29**		
**DIE Location at Second Surgery**									
Uterosacral ligament			0	1 (3.4%)	1 (3.4%)	1 (3.4%)	1 (3.4%)	7 (25.9%)	9 (18.4%)
Vagina			0	8 (27.6%)	5 (17.2%)	0	0	10 (37.0%)	23 (46.9%)
Intestine			1 (3.4%)	4 (13.8%)	4 (13.8%)	0	2 (6.9%)	12 (44.4%)	15 (30.6%)
Bladder			0	0	0	0	0	2 (7.4%)	3 (6.1%)
Others *			2 (6.9%)	1 (3.4%)	1 (3.4%)	0	1 (3.4%)	1 (3.7%)	5 (10.2%)

The 58 patients with OMA at first and second surgery are analyzed according to which side was involved; * umbilicus, appendix, inguina, round ligament of uterus. Note: The 105 patients with DIE at second surgery are analyzed according to the initial lesion subtype. For 29 of them it was DIE, for 27 SUP and for 49 OMA at the first surgery. Abbreviations: SUP, superficial peritoneal endometriosis; OMA, ovarian endometrioma; DIE, deep infiltrating endometriosis.

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
