# Peer review of "Recurrence Patterns after Surgery in Patients with Different Endometriosis Subtypes: A Long-Term Hospital-Based Cohort Study"

_jcm, 2020, doi:10.3390/jcm9020496_

Round 1

Reviewer 1 Report

Nirgianakis et al. aims to characterize the endometriosis lesion subtypes in first and subsequent surgeries. They also examined the lesions’ evolution and compared the time required for subsequent surgery based on the initial lesion subtype. They included surgically confirmed recurrent endometriosis, defined as subsequent surgery for endometriosis after a previous complete surgical excision of endometriotic lesions. They did not find a statistically significant difference in the time for recurrence of different severity types of endometriotic lesions. They also showed the progression of the disease was seen more in severe lesions compared to less severe forms.

This study emphasizes an important clinical question, which may help to understand the prognosis and progression of endometriosis. It is a very good study since it includes a long term follow up of surgically confirmed data on endometriosis lesions. The discussion and material methods are well written sections but the rest of the manuscript including abstract needs major revisions. The following are my suggestions.

1-Please rewrite the abstract and more importantly the results sections. The language and the sentences are very difficult to follow. The results should be clear. It needs significant English editing. The tables and figures are helpful but they need revisions as well. I suggest that more explanation is included in the tables and figures. Explanatory legends may help.

2-All tables including supplemental tables need revision and better labeling. Please fit the words appropriately to the tables. It may be helpful to explain what is being compared in legends. The results section is not clear and the tables are not helpful to follow the results.

3-The study population is well described and the limitations of the design are well addressed in the discussion section.

4-In regards to the comments on the possible pathogenesis of recurrence, I suggest including possible seeding during the surgery. This study discusses cases that had recurrence after a corrective surgery. Could this be one of the reasons for the recurrence seen in different locations than the primary lesion seen at the initial surgery?

5- This was already briefly included in the discussion section but any data on the other treatment modalities such as hormonal therapy etc will be very helpful in understanding the main question addressed. The nature of this study may not enable this part of the question but I think it is extremely essential if the authors can look into any medical therapy data. I am wondering if especially the group who was followed long term primarily within the study institution could have this information. This data will make the study stronger and it will be a critical information for the clinicians.

Reviewer 2 Report

The authors set out with a purpose of comparing endometriosis subtypes with respect to time to and subtype identified at subsequent surgery.  They were able to include 234 patients in the study.  Overall, this author found the results presentation difficult to follow and did not agree with interpretation of results.  Specific comments are as follows:

Abstract, Lines 26-28: “To summarize, increasing lesion subtype severity occurs in a significant proportion of patients at recurrence.”  What are the statistics that support this statement?  The information provided in Table 2 does not support this statement.

Results, lines 103-105: “In these patients, we observed a similar trend that showed the lesion subtypes at subsequent surgery tended towards a more severe subtype and in particular a DIE lesion.”  This statement is not supported by Supplemental Tables 1 and 2; The majority of SUP and OMA patients in each table recurred with the same subtype and no statistics are provided to support a trend.

Table 2, results line 115:  it should be specified the comparisons completed to generate the p-values; suggestion also to define OR

Discussion, lines 129-130: suggestion to edit this sentence to read: “In the present study, we demonstrate that the time to FIRST recurrent endometriosis surgery is independent from the endometriosis subtype observed at INITIAL surgery”.

Discussion, lines 132-133: “…there was an increased percentage of patients that present for more severe lesion subtypes, particularly DIE.”  It is unclear to this reviewer what is meant by “increased percentage”.  Comparing across tables, percentages of DIE in Table 2 for each subtype compared to that of Supplemental Tables 1 and 2 show no increases.  Comparing within tables, only a comparison of the column for SUP in Table 2 supports this claim; no comparisons for OMA in any table supports this claim.

Discussion lines 136-139: “The similarity in the time frames observed suggests that although some endometriosis subtypes are considered more severe and difficult to manage, they are not more aggressive in terms of recurrence and the need for subsequent surgical intervention.”  This reviewer does not agree with statement based on the findings of this manuscript; the analysis was inherently biased since the data was filtered to patients that recurred and no effort was made to determine whether there is a different in rate of recurrence or numbers of subsequent recurrences between subtypes.

Discussion lines 197-198: “However, we expect that possible incomplete endometriosis excision should have occurred homogeneously in each groups so that the findings should not be influenced”.  This reviewer does not agree with this assumption; authors should provide publications to support this and typo of “groups” should be edited.

Material and Methods, line 217: suggestion to replace “once” with “one”

Reviewer 3 Report

The study by Konstantinos Nirgianakis et al. entitled ,, Recurrence patterns after surgery in patients with different end

The study by Konstantinos Nirgianakis et al. entitled ,, Recurrence patterns after surgery in patients with different endometriosis subtypes: a long-term hospital-based cohort study” is a retrospective analysis of 3 groups of women with different type of endometriosis : superficial peritoneal endometriosis (SUP), ovarian endometrioma (OMA) and deep infiltrating endometriosis (DIE)

The authors tried to assess:

1.Time to recurrence and variation in endometriosis subtype between first and recurrent surgeries.

2.To find the association of the severity of the lesions in relation to the subtype in a subsequent operations.

They concluded that:

-Increasing lesion subtype severity occurs in a significant proportion of patients at recurrence.

-Time to recurrence is independent from the lesion subtype at first surgery.

The study is important as endometriosis is a cause of pain decrease quality of life and infertility.

I agree with the authors that one of the most important problem–also regarding that study is that we really could have doubts if the lesions were completely excised during first operation thus if the lesions found during second operation were the result of endometriosis persistence or endometriosis recurrence. I also agree that the limitation is they have no data regarding hormonal treatment applied after first operation. Finally that study is not free of selection bias which was described by authors. I wonder how many doctors in your clinic performed both the first and second /next operation(s).

However before publication some points need to be discussed.

The authors should state in the introduction what was the inclusion criteria in that study. The statement in abstract is not enough. According to the authors database: 322 women met the inclusion and exclusion criteria but only for 234 subtype at first surgery was confirmed -only that number of patients should be further analyzed . They stated that   88 women were exclude from further analysis. Thus I would like they recalculate their data for that proper number of patients – and exclude the women described as,, unknown subgroup” from table 1. The results section is not easy to read. The tables and graphs should be presented in manuscript in order of placement in text, thus tab 1 should be the first. Fig 1 a I fig 1 b should be removed and placed in supplement Do you have any data regarding how many of included women had MRI. Do you have data regarding diameter of endometrinoma. I also think that that in discussion section line 156-164 could be shorten. It is enough to write that ,,Whether endometriosis represents a progressive disease that continues to worsen over time is supported by some data and cited tchem.

Round 2

Reviewer 1 Report

Dear Authors,

I would like to thank the authors for their clarification and responses to the suggestions. The revised results section and tables in current form of the manuscript appears better to understand and follow. I suggest that the language and spelling are revised. 

Regards,

Reviewer 2 Report

The authors are still highlighting results that are not significant in the abstract and results sections without any acknowledgement that the results are not significant.  Specifically, they use revised results section 3.4 (lines 127-133) to highlight an increase in severity at recurrent surgery for patients that first present with SUP or OMA.  However, the statistics that accompany the increase from SUP to OMA are p=0.0036, OR 0.4.  In contrast, they completely ignore the statistics for OMA patients that recur with SUP p=0.0011, OR 0.28 and highlight a nonsignificant recurrence of OMA as DIE.  Likewise, there is no discussion of the proportion of DIE patients that recur with OMA (p=0.021, OR 0.42). 

In fact, the only statistically supported results with high odds ratios are the recurrence of SUP and OMA patients with the same subtype.  Furthermore, it is incorrect for the authors to make statements regarding increased severity for OMA and DIE subtypes without statistical support and without discussing decreased severity at recurrence.

Reviewer 3 Report

The authors gave the answers on my doubts.

Author Response

Point 1: The authors gave the answers on my doubts

Response 1: We thank you very much for the effort and time put into the review of the manuscript.